# Unveiling the Secrets of Oil Palm Genetics: A Look into Omics Research

**DOI:** 10.3390/ijms25168625

**Published:** 2024-08-07

**Authors:** Wen Xu, Jerome Jeyakumar John Martin, Xinyu Li, Xiaoyu Liu, Ruimin Zhang, Mingming Hou, Hongxing Cao, Shuanghong Cheng

**Affiliations:** 1National Key Laboratory for Tropical Crop Breeding, Chinese Academy of Tropical Agricultural Sciences, Haikou 571101, China; wenxu1638888@163.com (W.X.); jeromejeyakumarj@gmail.com (J.J.J.M.); lixinyu@catas.cn (X.L.); liuxy86@catas.cn (X.L.); zrm20010622@163.com (R.Z.); asd974692216@163.com (M.H.); 2Coconut Research Institute, Chinese Academy of Tropical Agricultural Sciences, Wenchang 571339, China; 3College of Tropical Crops, Yunnan Agricultural University, Pu’er 665000, China

**Keywords:** oil palm, genomics, transcriptomic, metabolomics, proteomics

## Abstract

Oil palm is a versatile oil crop with numerous applications. Significant progress has been made in applying histological techniques in oil palm research in recent years. Whole genome sequencing of oil palm has been carried out to explain the function and structure of the order genome, facilitating the development of molecular markers and the construction of genetic maps, which are crucial for studying important traits and genetic resources in oil palm. Transcriptomics provides a powerful tool for studying various aspects of plant biology, including abiotic and biotic stresses, fatty acid composition and accumulation, and sexual reproduction, while proteomics and metabolomics provide opportunities to study lipid synthesis and stress responses, regulate fatty acid composition based on different gene and metabolite levels, elucidate the physiological mechanisms in response to abiotic stresses, and explain intriguing biological processes in oil palm. This paper summarizes the current status of oil palm research from a multi-omics perspective and hopes to provide a reference for further in-depth research on oil palm.

## 1. Introduction

Oil palm (*Elaeis guineensis* Jacq) is known as the “oil king of the world”, and is the world’s highest oil-producing perennial tropical woody oil crop, with an average annual oil yield of 4270 kg/hm^2^ [1,2]. Although oil palm cultivation occupies only 5% of the global oil crop area, it supplies approximately 33% of vegetable oil and 45% of edible oil used worldwide [3]. This versatile crop is highly valued both economically and nutritionally [4,5,6]. The global demand for oil palm is expected to reach 240 million tonnes by 2025 [7]. A chart illustrating the recent supply of vegetable oils in the global market is presented in Figure 1 (https://www.statista.com/statistics/263937/vegetable-oils-consumption-worldwide/) (data sources refer to this web site: Vegetable oils consumption worldwide 2023/24 | Statista February 2024 (accessed on 22 March 2024)). Scientists and breeders are required to adopt newer tools and technologies to accelerate the development of high-yielding varieties that can adapt to challenging environments.

With the development and application of oil palm omics such as genomics, transcriptomics, metabolomics, and proteomics, researchers have carried out a wide range of studies to unravel the biological mechanisms and decipher the key genes in the development of oil palm fruits, including through the construction of genetic maps, pan-genomic studies, genome comparisons, and molecular marker development. At the transcriptome level, researchers have extensively utilized gene expression analysis to investigate the differential responses of sesame genotypes to biotic and abiotic stresses, as well as the functional roles of genes during tissue and organ development. Transcriptomics, proteomics, and metabolomics are used to determine the role of oil palm under different environmental conditions. Using multi-omics approaches, we identified candidate genes for essential agronomic variables attributed to oil palm development. These agronomic traits include high oil content, water and drought tolerance, disease resistance, cold and heat tolerance, and high yield, providing new avenues for the genetic development of oil palm.

The use of multi-omics has been helpful in the last few years to elucidate the responses to biotic and abiotic stresses in oil palm, as have methods for multidimensional studies. Figure 2 illustrates the different histological approaches employed in oil palm. This review of domestic and international research progress on oil palm genomics, transcriptomics, proteomics, and metabolomics highlights the various multi-omics approaches used in oil palm to identify genes supporting stress tolerance and molecular markers associated with important traits in oil palm, help in understanding protein–protein interactions and metabolic pathways for stress tolerance, and to offer future research directions. It also provides a vision for future research directions that can be used as information to improve oil palm quality at the molecular level.

## 2. Oil Palm Genomics

With the development of sequencing technology, researchers have been able to sequence the oil palm genome to facilitate the construction of molecular markers and genetic maps. This progress helps us to identify valuable genes related to the yield enhancement, fatty acid synthesis, oil accumulation, and resistance to abiotic stresses. Consequently, it aids in improving oil palm traits and helps us to understand the physiological mechanisms underlying important characteristics in oil palm.

### 2.1. Oil Palm Genome Sequencing

Singh conducted whole genome sequencing of oil palm using 454 pyrosequencing and Sanger BAC end sequencing technology [3]. The estimated genome size of oil palm is approximately 1.8 gigabases (Gb) [8,9]. Subsequently, the researchers used Illumina HighSeq 2500, Miseq, and Roche 454 sequencing technologies to randomly sequence the whole genome of the elite dura tree, assembling a total of 1.701 Gb, which covers 94.49% of the oil palm genome, and performing deep transcriptome sequencing of the major tissues and organs of the oil palm, annotating nearly 36,105 highly reliable oil palm genes [10]. Furthermore, a high linkage disequilibrium (LD) in Dura suggests that only a small fraction of SNPs covering the entire genome are needed for genome-wide association studies (GWAS) of important traits for marker-assisted selection (MAS) in oil palm compared to other agronomic crops, but the positional cloning of genes encoding important traits may be more challenging.

The availability of the whole genome sequence of the oil palm and the draft of the elite palm genome sequence represents a major breakthrough in the era of oil palm biotechnological genomics. In 2019, the first draft genome sequence of the *Ganoderma boninense*, the fungal pathogen responsible for basal stem rot (BSR) in oil palm, was published [11], which provided important genomic and expression profiling data for the study of oil synthesis and regulatory gene function and breeding for disease resistance and high yield.

In 2020, Ai-Ling Ong and colleagues improved the oil palm genome with a newly assembled genome called PMv6, with an N50 value of 83.1 Mb, which increased the scaffolding distribution of palm oil pseudochromosomes from 43% to 77% (1.2 Gb) and was published as an amplified version of the *E. guineensis* genome [12]. PMv6 was able to identify a more complete set of annotated genes and non-coding RNAs that may be responsible for phenotypic variations in QTLs following GWAS, providing a valuable resource for accelerating the genetic improvement of oil palm and investigating the mechanisms of phenotypic variation in important traits, including oil yield and quality, as well as disease resistance.

However, the early oil palm genome assemblies were fragmented, with only 43% of the sequences of the early assemblies located on pseudochromosomes. In 2022, a shared map (AM_EG5.1) was generated by integrating a vast array of SNP- and SSR-based genetic maps containing 828.243 Mb of genomic scaffolds. Notably, these scaffolds were successfully anchored to 16 pseudochromosomes, thereby accounting for a substantial 54% of the genome assembly [13]. The total length of the N50 scaffolds anchored to the pseudochromosomes increased by approximately 18% compared to the previous assembly. Additionally, 139 QTLs for important agronomic traits were successfully positioned on the new pseudochromosome map, based on the existing literature. The 3422 unique markers anchored to the genome sequence will be an important resource for genetic map construction. Genome sequencing has contributed to a deeper understanding of the genetic diversity of oil palm, providing numerous gene and variant profiles associated with important oil palm traits and environmental adaptations, and has propelled the advancement of oil palm research at the molecular level. In addition, an oil palm gene database has been established, including a taxonomic database (http://www.ncbi.nlm.nih.gov/Taxonomy/ (accessed on 22 March 2024)) of the National Centre for Biotechnology (NCBI, Bethesda, ML, USA) and PalmXplore access from the Malaysian Palm Oil Board (MPOB, Bandar Baru Bangi, Malaysia) ((http://palmxplore.mpob.gov.my/palmXplore (accessed on 22 March 2024)), providing information on genes related to important traits, such as fatty acid biosynthesis (FAB) and disease resistance [14]. Genomics and next-generation technologies are critical for ensuring a sustainable supply of oil palm with high yields and quality. The utilization of genomics technologies such as quantitative trait locus targeting, marker-assisted selection, and genomic selection can substantially increase yields in a shorter timeframe.

Chloroplasts have an important role in plant lipid synthesis. To gain a deeper understanding of the molecular basis of oil palm chloroplasts, the complete chloroplast (cp) genome was sequenced in 2012 using 454 pyrophosphate sequencing technology. The chloroplast genome is 156,973 base pairs long and contains 112 unique genes, including 79 protein-coding genes, 4 ribosomal RNA genes, and 29 transfer RNA (tRNA) genes [15]. In addition, sequencing of two parental *E. guineensis* individuals and their 23 F1 progeny using next-generation sequencing (NGS) revealed that the chloroplast genomes of 17 F1 progeny were identical to those of the parent. However, six individuals exhibited a single variant in the chloroplast genome sequence. Although the parent showed significant variation, all nucleotide variants were synonymous. This suggests that gene contents and sequences are highly conserved in the oil palm chloroplast genome. The maternal inheritance of the chloroplast genome in the F1 progeny was consistent, with a low likelihood of mutation across generations. These findings provide valuable insights into the inheritance patterns of the oil palm chloroplast genome, especially for crop scientists considering the use of the chloroplast genome for agronomic trait modification.

DNA methylation is a heritable epigenetic mechanism. The main methods for DNA methylation in oil palm include genome-wide bisulfite methylation sequencing (WGBS). Mantling in oil palm is an abnormal somatic clonal variant produced during the tissue culture process, and severe phenotypes can lead to loss of oil yield. E. Jaligot demonstrated a correlation between DNA hypomethylation and “MANTLED” somatic clonal variants in oil palm [16]. They sequenced and assembled 294,115 and 150,744 sequences from hypomethylated or gene-rich regions of the *E. guineensis* and *E. oleifera* genomes into overlapping clusters. Genes of interest were mined from gene models predicted by the assembled overlapping clusters, identifying 242 transcription factors, 65 resistance genes, and 14 miRNAs in oil palm [17]. Meilina Ong-Abdullah employed an epigenomic association study to identify the MANTLED locus in the African oil palm *(E. guineensis*). The study revealed that the mantled phenotype originates from a combination of two key epigenetic alterations: (1) changes in Karma methylation and (2) the loss of small RNAs during tissue culture. These findings provide novel insights into the molecular mechanisms underlying the mantled phenotype, highlighting the critical role of epigenetic modifications in shaping the phenotype of oil palm plants [18]. Norashikin Sarpan conducted a comprehensive whole-genome bisulfite sequencing analysis of seedlings, as well as normal and mantled clones. The results revealed a significant association between mantling and specific epigenetic features, including hypomethylation of CHG sites within the karma region of DEF1 and hypomethylation of hotspot regions on chromosomes 1, 2, 3, and 5. These findings suggest that alterations in DNA methylation patterns at these loci may contribute to the development of the mantled phenotype [19]. In another study, Siew-Eng Ooi’s research demonstrated that the methylation status of Karma-EgDEF1 was altered in clonal offspring (reclones) of phenotypically normal mother palms (clonal ortets), despite being normal in the original ortets. Notably, the study found that the reclones exhibited high coverage of Karma-EgDEF1 methylation, suggesting that this epigenetic change occurs during the clonal reproduction process. Furthermore, the identified differentially methylated regions (DMRs) have the potential to serve as valuable methylation markers for identifying high-risk clonal ortets, enabling the early screening and selection of healthy clones [20]. Changes in DNA methylation can enrich species diversity, and hypomethylated sequences provide an important resource for understanding the molecular mechanisms associated with important agronomic traits in oil palm.

### 2.2. Research on Molecular Markers for Important Traits in Oil Palm

The application of molecular markers in plant breeding is considered an effective method for large-scale screening of selective traits and to improve breeding efficiency. With the development of sequencing technology, molecular markers for important traits have been developed using DNA sequence-based tagged single nucleotide polymorphisms (SNPs), polymerase chain reaction (PCR)-based randomly amplified polymorphisms (RAPDs), restriction fragment length polymorphisms (RFLPs), and simple sequence repeats (SSRs) [21,22,23,24]. Amplified fragment length polymorphism (AFLP) methods combine PCR and RFLP markers [25]. Expressed sequence tags (EST) have facilitated comparative genomic analyses between oil palm (*Elaeis guineensis*) and other monocotyledonous and dicotyledonous plant species, enabling the identification of conserved and divergent genomic regions. Furthermore, ESTs have been utilized to develop gene-targeting markers for the construction of reference genetic maps, which provide a framework for mapping QTLs and identifying genes associated with desirable traits. Additionally, ESTs have been used to design and manufacture DNA arrays, which will enable high-throughput gene expression profiling and facilitate future studies on oil palm genomics, transcriptomics, and functional genomics [26]. Cleaved amplified polymorphism sequence (CAPS) molecular markers have been extensively validated as a robust and efficient tool for the detection of SNPs. The use of CAPS markers with novel SNPs has been shown to be able to target QTL for fruit traits [27]. Table 1 lists the molecular markers for important traits.

These techniques are being utilized to identify genes associated with traits of interest, offering new promise for the oil palm industry. For instance, SNPs identified through high-throughput sequencing technologies were utilized to investigate the genetic diversity of two *Elaeis* species, *E. oleifera* and *E. guineensis*, originating from diverse geographic locations, including Indonesia, Nigeria, Ghana, Angola, and Costa Rica [10]. Seven randomly amplified microsatellite markers (RAM) were used to characterize the genetic variability of 51 oil palm genotypes originating from the Congo, and it was found that the ACA and CGA primers yielded 241 alleles, with the number of polymorphic loci ranging from 46 to 14, respectively. The genetic diversity of the oil palm genotypes was found to be high, with a total heterozygosity of 0.64 and a significant proportion of polymorphic loci (89%). The RAM technique detected genetic variability in palm genotypes and showed high polymorphism and discriminatory sensitivity [28]. Genetic diversity was assessed using nine oil-palm-specific SSR primer pairs, which generated a total of 107 alleles from the two selected oil palm varieties, and cluster analysis revealed two major clusters, *E. oleifera* and *E. guineensis*, with a coefficient of similarity of 0.09, and, in the case of E. oleifera, the Chithara germplasm was grouped individually to indicate that its uniqueness was confirmed. Within the *E. guineensis* germplasm, G1 and G55 showed the greatest diversity [29]. Patterns of genetic diversity in wild *E. oleifera* germplasm and four cultivated *E. guineensis* palms were examined using RAD sequencing, a type of enzyme-based simplified genome sequencing. The study focused on a pan-genomic population from Honduras, Costa Rica, Panama, and Colombia. GWAS analyses on pan-genomic populations detected SNPs associated with fatty acid composition and yield traits. These genetic maps have been used to identify QTLs associated with oil yield and fatty acid composition, as well as traits like short trunk height (THT) and short frond length (FL). Molecular markers associated with the nutritional parameters, oil content, fruit colour [30], and drought tolerance of oil palm have also been discovered. Table 1 shows the molecular markers for important traits related to oil palm.

These results provide valuable insights that can guide the development of conservation strategies for the oil palm germplasm, as well as breeding programmes to obtain higher-yielding palm genotypes with superior quality and enhanced disease tolerance.

Furthermore, genotyping-by-sequencing (GBS) enables the rapid identification of SNPs as well as insertion and deletion (InDel) markers for the construction of genetic linkage maps [31], which facilitates the identification of significant DNA regions associated with the trait of interest via QTL analysis. Kwong and others developed the OP200K genotyping microarray, which has proven to be robust and consistent for applications in genetic diversity assessment, genome-wide association studies (GWAS), and genomic selection, and this SNP array has resulted in a significant improvement in current locus resolution, which has become the most densely genotyped array for oil palm [32].

Genotyping-by-sequencing (GBS)-based GWAS can be used for candidate gene detection [33]. A genome-wide association study of leaf spot resistance in 210 individual *tenera* palms from seven different (origin) crosses identified six SNP variants in oil palm significantly associated with loci for resistance to leaf spot [34]. Table 2 shows the functions of genome-wide association studies in oil palm.

**Table 1 ijms-25-08625-t001:** Molecular markers for important traits related to oil palm.

Molecular Markers	Important Traits	References
SNP, SSR	oil content	[35]
SSR	hull thickness	[36]
SNP	stem height	[37]
SSR	drought-tolerant	[38]
SSR;	drought-tolerant	[38]
SNP	drought-tolerant	[39]
SSR	illegitimate progenies	[40]
SSR	identification of QTLs in oil palm	[41]
SSR	cold tolerance	[42]
SSR	identification of QTLs associated with callogenesis and embryogenesis in oil palm	[43]
EST-SSR	fruit exocarp colour	[30]
RFLP	genotypic discrimination	[44]
SNP	fatty acid content	[45]
SNP	height of oil palm	[46]
Intron polymorphisms (IP)	oil palm fatty acid composition	[47]
SSR	genetic diversity	[48,49,50]
SSR	sex ratio	[51]
SNP	vitamin E biosynthesis	[52]
SNP-CAPS	fruit traits	[53]

### 2.3. Gene Resource Mining and Function Analysis of Important Traits in Oil Palm

Plant growth and development are highly complex processes involving the formation of various tissues, organs, colours, and qualities. Therefore, plants require a large number of genes and metabolites to elucidate their growth and development as well as regulatory mechanisms. Genomics has been used to explore the genes corresponding to important plant traits and to mine genetic resources to improve plant quality. Figure 3 shows the breeding technology for oil palm.

In 2013, Singh and others constructed a high-quality genome of the oil palm and found that the *SHELL* gene regulates oil palm yield. It was later determined that haploinsufficiency characterizing *SHELL* genes affects morphological variability in oil palm fruits [62]. In 2014, the *VIRESCENS* gene was identified to control fruit colour [63]. A large number of studies have been conducted to elucidate the role of different gene families in oil palm. Some representative studies are listed in Table 3.

Fatty acid synthesis is a critical process in oil palm, as it is responsible for the production of the unsaturated fatty acids that are the primary components of oil palm oil. A recent study identified 42 key genes involved in oil palm fatty acid biosynthesis through a combination of comparative genomics analysis, the characterization of conserved structural domains and active sites, and expression analyses. Additionally, 210 candidate resistance genes in six categories were identified [64]. Ying isolated a *GDSL* esterase/lipase gene from oil palm significantly correlated with oil content and named it *EgGDSL*. Notably, their research revealed that the transcript abundance of EgGDSL is strongly associated with the extent of oil accumulation, suggesting a critical role for this gene in modulating oil production in oil palm [65]. Diacylglycerol acyltransferase type 2 (DGAT2), a pivotal enzyme in the oil palm genome, plays a vital role in the biosynthesis of triacylglycerol (TAG), a fundamental component of palm oil. As a key catalyst in the final step of TAG assembly, DGAT2 facilitates the esterification of diacylglycerol with acyl-CoA, thereby regulating the production of TAG in oil palm [66]. The four *DGAT* taxa of the *DGAT* gene (i.e., *DGAT1*, *DGAT2*, *DGAT3*, and *WS/DGAT*) have distinct physiological roles, particularly in developmental processes related to reproduction (e.g., flowering) and fruit/seed formation, especially in mesocarp and endosperm tissues [67]. In another study, a yeast one-hybrid assay was utilized to screen the oil palm MADS box gene EgMADS21, using the EgDGAT2 promoter sequence as a molecular bait. The results demonstrated that EgMADS21 exerts regulatory control over EgDGAT2 expression, thereby influencing the accumulation of fatty acids in the pericarp of oil palm [68].

In terms of organ development, the early nodulin 93 protein gene (*ENOD93*) gene plays a crucial role in the inducing of somatic embryogenesis and is highly expressed in leaf explants with embryonic potential [69,70]. Additionally, the expression patterns of miR156/529/535-SQUAMOSA promoter binding protein-like (*SPL*) gene were found to distinguish between early-developing male and female oil palm inflorescences [71]. A thorough examination of the oil palm genome identified 24 EgSPL genes, with 14 of these genes found to be regulated by miR156. Most of the *EgSPL* genes were actively expressed in the reproductive structures of oil palm plants, specifically in the female and male flower clusters located above the trophic tissues, suggesting that *EgSPL* genes have an important role in inflorescence development [72].

For important traits of oil palm, such as oil yield, fatty acid synthesis, sex ratio, and resistance, which are closely related to people’s lives, the mining of genes associated with these traits is beneficial for oil palm breeding and production. Using genomic tools to improve oil palm is an effective approach to achieve sustainably high yield and quality in oil palm [2].

**Table 3 ijms-25-08625-t003:** A representative study of oil palm fatty acid composition and gene families involved in response to biotic and abiotic stresses.

Gene Family	Function	Reference
Abiotic stress		
*CBF*	Increased tolerance to low temperatures	[73]
*ABA*	Enhance resistance to drought	[74]
*bZIP*	Highly expressed under low temperature, salinity, and drought stress conditions	[75]
*AP2/ERF*	Increased tolerance to salt, low temperatures, and drought	[76]
*HRE2(* *ERF-VII)*	*HRE2* responds to flood-induced hypoxia	[77]
*MYB*	Increased tolerance to stresses such as salinity, cold, and drought	[78]
*WRKY*	Involved in abiotic stress responses such as drought, salinity, and heat	[79]
*EgSPCH(bHLH)*	*EgSPCH* produces more pores in response to salt stress	[80]
*ARF*	*EgARF* responds to abiotic stresses (cold, drought, and salt stress)	[81]
*CAT*	*EgCAT* enhances tolerance to low temperature, drought, and salt stress	[82]
Biotic stress		
*CNL R gene*	Expression in the early stages of oil palm defence mechanisms	[83]
*AGLU1* *RCH10*	Increasing oil palm resistance towards *G*. *boninense*	[84]
*LCC*	*EgLCC24* gene detects oil palm seedlings resistant to *G. boninense*	[85]
Fatty acids		
*MADS-box*	Fruit development and oil accumulation	[86]
*MAD*	Mediating medium-chain fatty acid (*MCFA*) anabolism in oil palm mesocarp	[87]
*GDSL*	Enhanced lipid content	[65]
*ACBP*	Lipid accumulation	[88]
Others		
*CAD*	Lignin biosynthesis	[89]

## 3. Oil Palm Transcriptomics

Transcriptomics is the essential link between the genetic information of genes and biological functions, making it one of the most active disciplines in the post-genomics era [90]. Current oil palm transcriptomics research mainly relies on high-throughput RNA sequencing (RNA-seq), PCR technology, and cDNA, utilizing platforms like Illumina NextSeq 500. Luo constructed ArecaceaeMDB (http://arecaceae-gdb.com (accessed on 22 June 2023)), the first comprehensive multi-omics analysis platform for the Palmaceae family, provides users with multi-omics data containing genomes, variomes, transcriptomes, and metabolomes and embeds useful bioinformatic tools such as Blast, GO Enrichment and KEGG Enrichment [91]. 

The transcriptome can reveal the molecular mechanisms and gene expression information of organisms responding to environmental stress. Abiotic stresses in oil palm include waterlogging, drought, cold, salt, etc., while biotic stresses are mainly from bacteria, pests, etc., which alter gene expression. Table 4 shows the transcriptomic studies analyzing stress responses in oil palm. These theoretical studies are important for breeding new oil palm varieties with high salt and heavy metal tolerance, laying a foundation for subsequent research in oil palm genetics and breeding.

The transcriptome can provide information on the expression of genes regulating the synthesis and accumulation of free fatty acids, revealing mechanisms controlling differences in oil content, fatty acid (FA) composition and metabolism [92] as well as genes related to fatty acid synthesis [93,94]. Studies on the gene expression profiles of oil palms from six fruit developmental stages in three different oil-producing groups (low, medium, and high) have demonstrated that the transcriptomic profiles of each developmental stage are unique [95]. In addition, the genes of the pericarp-specific *MADS-boxes* at different developmental stages in oil palm suggest that these genes may play important roles in the process of fruit cell division and metabolite accumulation, and thus become important targets for the study of oil palm fruit development and oil accumulation [86], revealing the biological mechanism of the oil palm’s high oil production [96].

The examination of gene expression across diverse tissues and organs at various developmental stages is paramount to gaining a comprehensive understanding of plant development and regulatory mechanisms, as well as to enhancing plant transformation and utilization. The transcriptome of unpollinated oil palm pistils and pistils at 2 h, 4 h, 12 h, 24 h, and 48 h post-pollination reveals dynamic gene expression profiles that drive the pollination and fertilization process. The differentially expressed genes were found to be predominately associated with energy metabolism and hormone signalling pathways, providing a theoretical basis for the sexual reproduction of oil palm [97]. These studies have provided resources for a deeper understanding of the genetic characteristics of oil palm and the specificity of different tissues, and have laid the foundation for the structural improvement of oil palm.

The transcriptional applications mentioned above are mainly related to coding proteins. Recent studies have focused on the identification and characterization of these ncRNAs in oil palm. For example, the deep sequencing of small non-coding RNAs from five pericarp developmental stages identified 452 microRNAs (miRNAs). Among these, 22 conserved miRNAs and 14 novel miRNAs were found to potentially play a role in FA metabolic pathways [98]. Another study systematically characterized the expression patterns of the lncRNA motif and its target genes using strand-specific RNA-seq data and 18 published oil palm transcriptomes. Overall, 6882 lncRNA SNPs were detected, of which 28 SNPs were associated with variations in fatty acid content and 7 lncRNAs showed a positive correlation with the expression patterns of genes involved in the fatty acid from-scratch synthesis pathway [99]. In addition, MicroRNAs (miRNAs) play a crucial role in the transcriptional and post-transcriptional regulation of gene expression and are important molecules for plants to cope with abiotic stress.

Transcriptomics technology reveals the gene expression mechanism of oil palm under various stresses such as temperature, waterlogging, drought, high salt, and disease. Additionally, it provides insight into the synthesis mechanisms of fatty acids and gene expression patterns during oil palm fruit development. Furthermore, investigating non-coding RNAs in oil palm can facilitate molecular breeding efforts.

**Table 4 ijms-25-08625-t004:** Transcriptomic studies on analyzing stress responses in oil palm.

Trait for Transcriptome Analysis	Transcription Factor	Summary	Reference
Somatic embryogenesis	*FTIP*, *FRIGIDA-LIKE*, and *NF-YA*,	The reproductive time of plants may be related to somatic embryogenesis potential	[100]
Drought	*MYBs*, *HOXs*, *NF-Ys*	A potential miRNA target gene for oil palm under water deficit conditions	[101]
Pi-deficiency	*PHL7*, *NIP6–1* and 14–3-3	Involved in the lack of regulation and domestication of oil palm Pi	[102]
Waterlogging	*LBD41*, *HRA1*, *HRE2*	Respond to waterlogging	[77]
High somatic embryogenesis rate	*LE*, *DDX28*, *HD-ZIP*, *NPF*	The physiological state related to somatic embryogenesis potential was revealed	[103]
Bud Rot Disease	*Clon34* and *Clon57*	Control pathogen	[104]
Defoliation stress	/	Elucidating the mechanism of oil palm sex determination.	[105]
Bagworm stress	*P450*, *GST* and *CCE* *AChE*, *nAChR*, *ACCase*, *GABA*, *RyR*	Insecticide targets for detoxification and modification potential	[106]
Aluminium stress	*DREB1F*, *NAC*	Induces the expression of internal detoxification enzymes	[107]
Water deprivation	*NF-YA3*, *HOX32*, *GRF1*	It plays a certain role in the response of oil palm seedlings to drought and salt stress.	[108]
Salt stress	*MYC*, *G-Box*, *ABRE*, *TATA-box*	Using identified genes to try to mitigate salt stress	[109]
*G. boninense* infection	target gene #5, #13, #16, #6 and #15, #1, #9, #12, and #14	It may be related to early infection reaction to *G. boninense*	[110]

## 4. Oil Palm Proteomics

Transcriptomics alone is not sufficient to capture the complex dynamics of plant systems biology. To gain a more comprehensive understanding, it is essential to analyze the composition, expression levels, and post-translational modification states of proteins, and we can understand the interactions and relationships between proteins and explain the phenomena of life at a holistic and functional level. In recent years, peptide-centred proteomics such as matrix-assisted laser ionization mass spectrometry (MALDI)–time-of-flight mass spectrometry (TOF)/TOF, difference gel electrophoresis (DIGE), GeLC-MS/MS, and two-dimensional (2D) LC-MS/MS have been used to study proteins involved in the regulation of oil palm fatty acid biosynthesis and other biological processes. 

Proteins are the executors of gene functions and can provide a direct basis for understanding the nature of the stress response. Moreover, due to the regulation of post-translational modifications of proteins such as variable splicing and phosphorylation of mRNAs, the proteome is more complex and diverse than the transcriptome and genome, and its scope of study is wider. Therefore, explaining how oil palm responds to stresses such as cold, drought, high salt, and disease stress at the protein level provides a theoretical foundation for oil palm growth and development. Proteomics technology has been actively used to study the interaction between oil palm and the pathogenic fungi, *G*. *boninense.* Additionally, proteomic analysis of three oil palm varieties using isobaric tags for relative and absolute quantitation (iTRAQ) technology have identified proteins implicated in the response to diverse stresses, including abiotic, biotic, oxidative, and heat shock, as well as those involved in photosynthesis and respiration. The increase in stress-responsive proteins and the decrease in the photosynthesis-associated proteins provided insights into the molecular mechanisms of adaptation to low-temperature stress in oil palm varieties in China [111].

Protein changes during somatic embryogenesis involve cell differentiation. Proteomics reveals the molecular mechanisms of plant embryonic development [112]. The proteins that are differentially expressed during this process are identified at early stages of embryogenesis and are specific to particular stages. Gel quantitative proteomics has been utilized in other studies to understand the biological mechanisms underlying low-level embryogenesis. Tan Hooi Sin identified three proteins that could serve as potential biomarkers for the proliferation of healing tissue in oil palms, namely acid prolyl glucose isomerase, L-ascorbate peroxidase, and superoxide dismutase [113]. Further research indicated that antioxidant and cytokinetic proteins, as well as proteins involved in the ubiquitination pathway, may also be potential biomarkers for the acquisition of embryogenic capacity [114]. Moreover, E3 ubiquitin-protein ligase and sister chromatid-monomer cohesion PDS5 are specifically expressed only in somatic embryos and plantlets, and can be used as protein biomarkers to determine the maturation stage of oil palm somatic embryos [115].

Adjusting the fatty acid composition of the oil palm to meet the needs of the food and oleochemical industries can further enhance its performance. Proteomic analyses provide insights into the molecular mechanisms of fruit metabolism and oil accumulation. Differential gel electrophoresis (DIGE) can distinguish key proteins associated with different agronomic traits and control mechanisms related to lipid biosynthesis. The results imply that the orchestration of lipid biosynthesis involves a complex interplay of multiple metabolic pathways. The researchers provided strong support using shotgun quantitative proteomics techniques and identified several proteins from other metabolic pathways (e.g., glycolysis) that accumulate to varying degrees and may contribute to the regulation of fatty acid biosynthesis [116].

In a post-translational modification (PTM) study, researchers used target peptide fragments corresponding to fatty acid proteins for selected reaction monitoring (SRM) and found that an increase in oxidative phosphorylation activity was critical for an increase in oil production [117]. There were differences in the expression of essential proteins involved in oil palm fruits with different unsaturated lipid contents. The dynamics of the oil palm fruit proteome during early developmental stages is predominantly characterized by intense activity in fundamental biological processes, including primary metabolic pathways, energy production and utilization, stress response mechanisms, cell structural organization, and protein synthesis and degradation dynamics [118]. Using 2DE to analyze the protein profiles of oil palm fruits at different stages of ripening, 68 differentially expressed protein spots were successfully identified. Proteomic analysis of oil palm fruit development reveals that proteins involved in lipid biosynthesis, energy production, secondary metabolite synthesis, and amino acid metabolism undergo the most significant changes, providing valuable insights into the regulatory mechanisms controlling protein expression during fruit ripening and lipid accumulation [119].

Elucidating complex signalling pathways is a challenging and labour-intensive task. Consequently, integrating multiple histological methods, including proteomics, to detect dynamic changes in oil palm trees is crucial and is being continuously refined to enhance oil palm tree understanding.

## 5. Oil Palm Metabolomics

Metabolomics can reveal physiological processes by examining local responses to stimuli and pathogenesis. The metabolome is influenced by both genetic and ecological changes. In oil palm, metabolomics is commonly detected using gas chromatography–mass spectrometry (GS-MS), liquid chromatography–mass spectrometry (LC-MS), and nuclear magnetic resonance (NMR). A metabolomics-based review of advances in the study of important tropical plants introduced analytical sampling and sample preparation tools such as chromatography and mass spectrometry, and also highlighted a new metabolomics methodology that broadly targets metabolomics, summarizing detected metabolites in oil palm, including lipids, amino acids, organic acids, and amines, and combining them with multi-omics.

Metabolomics can visualize the metabolic mechanisms of stress resistance in oil palm. Using an untargeted metabolomics approach to study differentially expressed metabolites in healthy oil palms and those exhibiting yellowing symptoms, MS-based metabolomics analysis detected over 50 metabolites in oil palm leaf samples and identified nine differentially expressed metabolites. Among these, glycerophosphatidylcholine and 1,2-dihexanoyl-sn-glycerol-3-phosphate ethanolamine were considered potential biomarkers [120]. The most affected pathways under drought stress were starch and sucrose metabolism, glyoxylate and dicarboxylic acid metabolism, alanine, aspartate and glutamate metabolism, arginine and proline metabolism, and glycine, serine and threonine metabolism [121]. 

In another study, hydroponics maintained both ideal and non-optimal Pi supply conditions for 14 days. Metabolite assays were performed on leaves and roots using the GC-MS technique, which showed that storage lipids metabolize cellular energy in the absence of Pi [122]. The pathogenicity of monokaryotic (monokaryon) and dikaryotic (dikaryon) mycelia of the oil palm *G*. *boninense* pathogen was compared using a metabolomics approach, revealing that the monokaryon produced a lower amount of fungal metabolites compared to the dikaryon, indicating a reduced likelihood of plant infection [123]. Notably, the identification of crucial genetic elements, comprising genes, gene products, and metabolic pathways, has been accomplished, and these are implicated in the defence mechanisms of oil palm against Ganoderma boninense infection. Furthermore, these metabolites have been recognized as having potential utility as biomarkers for the early detection of this disease [124].

Additionally, nucleosides were found to be at higher concentrations during lipid biosynthesis, while metabolites involved in the tricarboxylic acid cycle were found to be at higher levels during early fruit development [125]. Furthermore, a metabolomics-based profiling of oil palm leaves was conducted, which revealed that oil palm spear leaf extracts comprise a diverse array of compounds, including three amines, twenty amino acids, and six organic acids. This comprehensive characterization provides a foundation for investigating the relationship between these metabolites and oil palm agronomic traits [126].

The regulation of plant metabolites cannot be simply inferred from genetic data. The relationship between metabolites and molecular functions is ambiguous and requires comparisons between gene expression QTLs (eQTLS) for biosynthetic genes and metabolite QTLs. Indeed, elucidating the intricacies of metabolic pathways requires a multifaceted approach, incorporating a range of experiments designed to evaluate biochemical reactions and fluxes within the pathway under diverse conditions.

Metabolomics intuitively reveals the metabolic regulation mechanism and fatty acid synthesis mechanism of oil palm in response to stresses such as BRS and temperature. The convergence of genomic principles and cutting-edge analytical methods, including genome, transcriptome, and proteome analysis, offers a powerful framework for elucidating the intricate mechanisms underlying biological processes. By leveraging these integrated approaches, researchers can uncover novel insights into the genetic and molecular basis of complex traits, ultimately enabling the development of improved agronomic characteristics and more resilient crop varieties. Meanwhile, the biological functions of primary and secondary metabolites in oil palm growth, development, and defence require in-depth exploration.

## 6. Conclusions and Outlook

With the rapid development of sequencing technology, research related to oil palm genomics is increasing. The completion of the oil palm genome assembly marks a significant advancement in oil palm research and introduces new opportunities for other histological research developments. The genomic sequences of specific oil palm cultivars have been comprehensively elucidated, and high-density genetic linkage maps have been meticulously constructed, thereby furnishing a robust framework for deciphering the intricate genetic underpinnings of this economically significant crop. Methylation studies can better link genotypes and epigenetic genotypes, providing a more comprehensive understanding of oil palm genetics.

Genomics research has facilitated the rapid development of the transcriptome, proteome, and metabolome, becoming a primary approach in unravelling the genetic diversity of oil palm. This includes the study of fatty acid synthesis, plant development, tissue and organ differentiation, and the potential mechanisms of plant adaptations to biotic and abiotic stresses. Utilizing this valuable information can facilitate the selection of planting materials with desirable traits such as drought tolerance, high nutritional status, salinity, drought, disease and cold resistance, and high productivity.

The opportunity to breed these varieties through experimentation at the molecular level can be applied to practical production, enabling oil palm to overcome its current productivity challenges. In addition, the synergistic amalgamation of multi-omics methodologies precipitates a more nuanced understanding of the intricate mechanisms underlying the complex phenotypic manifestations of numerous agriculturally pertinent traits. As a consequence, this paradigm enables the large-scale identification of a vast array of genes and signals implicated in biological processes, thereby substantially enhancing the capacity to optimize and refine oil palm cultivation practises. Looking ahead to the future of oil palm genomics, we will focus on addressing the current research gaps in China and building upon the existing hotspots explored by researchers worldwide. Our future research directions will include the following. 

(1)Constructing more high-quality oil palm transcriptome databases, further studying the biological functions of non-coding RNAs, and conducting research on the molecular mechanisms of sexual reproduction in oil palm.(2)Strengthening the research on oil palm growth, development, and defence metabolism. Targeted metabolomics is crucial for an understanding of specific metabolites or changes in metabolic pathways under different environmental conditions. By combining targeted and non-targeted metabolomics, we can strengthen the research on the identification and interactions between primary and secondary metabolites, the regulation mechanisms of metabolites in oil palm growth and development, and the role of secondary metabolites in the defence mechanism.(3)Developing and applying low-cost and rapid techniques for oil palm genome determination, and strengthening research on genetic diversity and epigenetic modification in response to stress in oil palm species.(4)Exploring the responses of oil palm to multiple stresses and the genetic mechanism of complex agronomic traits by combining multi-omics technologies and integrating genomics, transcriptomics, lipidomics, metabolomics, and proteomics, so that we can identify regulatory genes of complex agronomic traits and to clarify the mechanism of action and metabolic pathways, so as to provide a theoretical basis and genetic resources for modern molecular breeding.

By leveraging multi-omics technology, we can uncover the intricate relationships between oil palm’s genetic makeup, gene expression, and metabolic responses to multiple stresses, ultimately informing the development of more resilient and productive oil palm varieties.

## Figures and Tables

**Figure 1 ijms-25-08625-f001:**
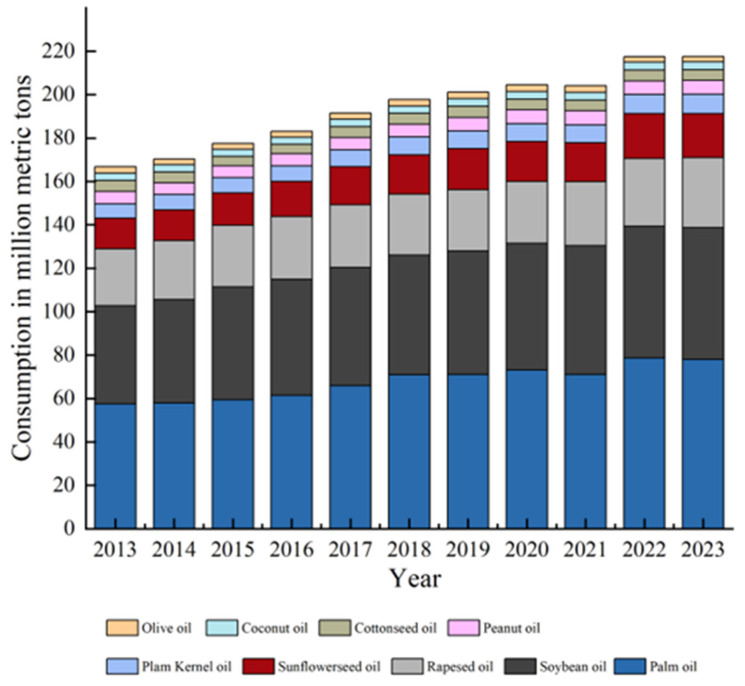
World supply of vegetable oils.

**Figure 2 ijms-25-08625-f002:**
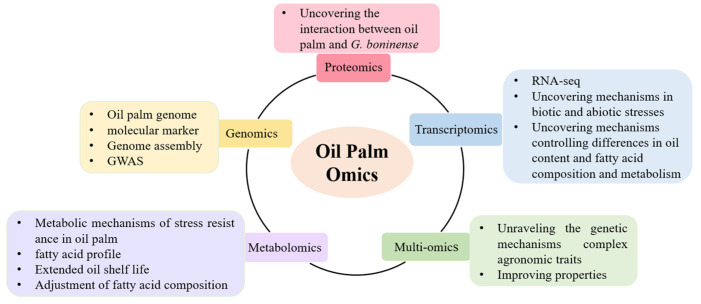
The schematic representation of various omics approaches deployed in oil palm.

**Figure 3 ijms-25-08625-f003:**
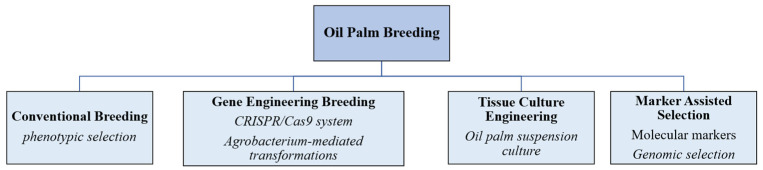
Oil palm breeding methods.

**Table 2 ijms-25-08625-t002:** Functions of genome-wide association studies in oil palm.

Traits	Number of Accessions	Chromosomes	Number of QTLs	Number of Candidate Genes	Reference
compact fruit bunch	422	4, 8, 11	4	/	[54]
oil content	196	/	33	55	[55]
fatty acid composition; nutrient content	210	2, 13	6	3	[34]
oil content	471	3, 5, 10, 13, 15	/	/	[56]
disease resistance	96	6, 7, 9	5	/	[57]
leaf area size; high yield;	115	/	/	5	[58]
trunk height	132	5, 6	/	/	[59]
stem height	96	1, 2, 4, 14, 16	7	/	[60]
high oil production	310	1, 4, 7, 10, 11, 12, 15	43	/	[61]

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
