# Peer review of "Unveiling the Secrets of Oil Palm Genetics: A Look into Omics Research"

_ijms, 2024, doi:10.3390/ijms25168625_

Round 1

Reviewer 1 Report

Comments and Suggestions for Authors

This is a comprehensive review of omics studies of oil palm. This review summarizes papers devoted to oil palm researches for a wide period and generalizes findings obtained from genome, transcriptome, proteome, and metabolome. The review will be of interest to researchers in broad fields. Although the manuscript is well-written, there are concerns that should be addressed as described below.

There are typographical and grammatical errors throughout the manusctipr and these errors are needed to be edited.

Table 2: A horizontal line as a border line should be inserted below the top row. The left column (traits) is too complicated to understand. 

Table 3: The middle coulm (Function) is too complicated. Is it possible to show concisey?

Table 4: The middle coulm (Summary) is too complicated. Is it possible to show concisey?

L34-36: References for the data on oil palm market shown in Figure 1 should be shown.

Figure 2: There is no description/explanation on lipidomic in the text. Because fatty acids are also target compounds in metabolomics, I would recommend that lipidomic and metabolomics are combined in one category.

L55: The term genomics/multi-omics is confusing. There are cases in which genomics is included in Multi-omics.  

L204: The abbreviation for single nucleotide polymorphisms (SNP) is defined at first mention in the text. Similar cases for other terms are seen throughout the manuscript.

Figure 3: Overall, it is difficult to understand. Does top panel mean a timeline showing historical events of important techniques/methods? What is the meaning of different colors (blue, yellow, and gray)?

3. Oil Palm Transcriptomic (L267-360): Overall, there are a number of redundant/overlapping phrases and expression throughout the section. This section should be written concisely.

L376-382: References for these findings should be shown.

L408-412: References for these findings should be shown.

Comments on the Quality of English Language

English of the manuscript should be edited.

Reviewer 2 Report

Comments and Suggestions for Authors

Overall, authors summerized the important achievements of oil palm multi-omics studies, and I believe it will definitely benefits oil palm reserach community for reference. For the manuscript, the only suggestion I have is that authors need to add some contents on proteomics and metabolomics in the conclusion to follow the main part. 

Some minor points for authors:

1. Line 54, "national" is included in the "international", here "national" means "Chinese"?

2. Author need to pay attention on tense of the verbs, such as like line 75, "be" should be in past tense. 

3. Line 177, line 186, line 187, E.oleifera should be in italic.

4 Line 346, de novo should be in italic.

5 Line 390, one more blank between “proliferation” and "of".

6. Line 235, you need press backspace key. 

Do some proofreading later please. 

Comments on the Quality of English Language

Acceptable. A little bit minor revisions on grammars needed.  

Round 2

Reviewer 1 Report

Comments and Suggestions for Authors

The manuscript has been improved.